# Inhalative Treatment of Laryngitis Sicca in Patients with Sjögren’s Syndrome—A Pilot Study

**DOI:** 10.3390/jcm11041081

**Published:** 2022-02-18

**Authors:** Benedikt Hofauer, Lara Kirschstein, Simone Graf, Ulrich Strassen, Felix Johnson, Zhaojun Zhu, Andreas Knopf

**Affiliations:** 1Department of Otorhinolaryngology/Head and Neck Surgery, Klinikum Rechts der Isar, Technical University of Munich, 80538 Munich, Germany; larakir@web.de (L.K.); graf.simone@tum.de (S.G.); ustrassen@protonmail.com (U.S.); fallendorff@gmail.com (F.J.); yaya.zhu@gmx.de (Z.Z.); 2Department of Otorhinolaryngology/Head and Neck Surgery, University Medical Center Freiburg, University of Freiburg, 79106 Freiburg, Germany; andreas.knopf@uniklinik-freiburg.de

**Keywords:** Sjögren’s syndrome, laryngitis sicca, inhalation, liposomes, ESSPRI, ESSDAI

## Abstract

Xerostomia and keratoconjunctivitis sicca are the main symptoms of Sjögren’s syndrome. Often patients also suffer from laryngeal complaints, but there is a lack of specific treatment options. The aim of this study was to evaluate the effect of a liposomal inhalation therapy. Patients with Sjögren’s syndrome were included and received a two-month period of liposomal inhalation therapy. The effect was evaluated by standardized questionnaires (patient-reported indices) and measurement of unstimulated whole salivary flow and glandular stiffness. Forty-five patients were included in this study. A comparison of baseline and therapeutic values demonstrated a significant improvement of the EULAR Sjögren’s syndrome patient reported index (ESSPRI) with a baseline of 5.0 ± 2.1 and a therapeutic value of 4.1 ± 2.4 (*p* = 0.012). This improvement was mainly based on the item on dryness within this score. Overall, the therapy was well tolerated. In conclusion, an inhalative application of liposomes had a beneficial effect on the reported dryness in patients with Sjögren’s syndrome. A first insight into the effect of inhalation therapy on laryngeal symptoms could thus be obtained and at the same time the basis was created on which case calculations can be carried out in the future.

## 1. Introduction

Sjögren’s syndrome is a chronic, systemic autoimmune disorder which is characterized by lymphocytic infiltrations of the lacrimal and salivary glands [1]. This chronic inflammation causes progressive destruction of the glands and leads to xerostomia and keratoconjunctivitis sicca [2]. The reduced production of saliva can result in numerous other complaints in the area of the upper respiratory and digestive tract, such as periodontal disease, loss of teeth and oral candidiasis [3,4,5,6,7]. In addition, recent reports have shown consistently, that the hyposalivation in patients with Sjögren’s syndrome also leads to a laryngeal impairment, resulting in dysphonia and dysphagia [8,9].

Treatment options for Sjögren’s syndrome have not changed significantly over the past few decades. While the sicca symptoms are primarily treated symptomatically, various immunomodulating drugs are available for systemic manifestations. Under the leadership of the EULAR (European League Against Rheumatism) Sjögren’s Syndrome Task Force Group, recommendations on the management of Sjögren’s syndrome were published last year. The recommendations for the treatment of oral dryness aim to reduce the discomfort within the oral cavity and in the area of the pharynx while the laryngeal impairment is not addressed separately [10].

For a number of years, a topical treatment option that has been shown to be helpful in the treatment of dry mouth in patients with Sjogren’s syndrome, as an oral spray, has also become available in the form of an inhalation solution [11]. This topical treatment consists of liposomes, which contain phospholipids and enables, in addition to a humidifying effect, a restoration of the surfactant film. A recently published, randomized controlled trial on the effect of a liposomal inhalation compared to a standard physiologic saline inhalation early after tracheostomy was demonstrated to be beneficial with regard to the inflammatory parameter [12].

For this reason, the aim of this study was to evaluate the effect of liposomal inhalation therapy on laryngeal symptoms in patients with Sjögren’s syndrome. Since no data are available on the effect of this kind of therapy in such a collective of patients yet, another aim of this study is to provide information on the effect of this therapy to enable future case calculations.

## 2. Study Design and Methods

### 2.1. Study Population

Patients diagnosed with primary Sjögren’s syndrome based on the ACR-EULAR classification criteria were invited to take part in this examination independent of specific clinical symptoms [13]. The study was conducted at the Department of Otorhinolaryngology/Head and Neck Surgery at the Klinikum rechts der Isar, Technical University of Munich, Germany. The study protocol was in accordance with the Declaration of Helsinki. The Institutional Review Board of the Medical Faculty, Technical University of Munich, reviewed and approved the protocol. Written informed consent was obtained from all participants prior to the beginning of the intervention. Patients with past head and neck radiation, Hepatitis C infection, AIDS, pre-existing lymphoma, sarcoidosis, graft versus host disease, recent use of anticholinergic drugs or known allergies for ingredients of the inhalation had been excluded. Subjective symptoms (xerostomia, keratoconjunctivitis sicca, parotidomegaly) were evaluated with visual analogue scales. Unstimulated salivary flow (UWSF) and Schirmer’s test were conducted to evaluate salivary and lachrymal gland function. Antibodies to Ro (SS-A) and La (SS-B) antigens were detected. If required for the classification, minor salivary gland biopsy was performed. The EULAR Sjögren’s syndrome (SS) disease activity index (ESSDAI) was applied to measure disease activity [14,15].

### 2.2. Outcome Parameters

Different parameters were evaluated prior to and after the inhalative treatment:-ESSPRI: The EULAR SS patient reported index was completed by the patients. This index contains the variables of pain, fatigue and dryness and are scored by the patients in between a range of 0 and 10, the final score is the mean of the three single items [15].-EORTC QLQ H&N 35: The quality of life questionnaire of the European Organization for Research and Treatment of Cancer incorporates nine multi-item scales, five functional scales (physical, role, cognitive, emotional and social), three symptom scales (fatigue, pain and nausea/vomiting) and a global health and quality of life scale [16].-ADI: The Anderson Dysphagia Inventory consists of 20 questions from the global, emotional, functional and physical domains. Each question can be scored 1 to 5 points. A value of less than 55 is considered “highly noticeable”, 55–70 as “rather noticeable” and greater than 70 “not noticeable” [17].-VHI: The Voice Handicap Index measures voice-related impairment of the quality of life in the functional, physical and emotional dimension. VHI values of 0–11 are classified as grade 0 (almost certainly no noticeable grade of suffering), values of 12–28 a grade 1 (more likely unnoticeable than conspicuous grade of suffering), values of 29–56 as grade 2 (more probably noticeable suffering than not) and values of 57–120 as grade 3 (certainly noticeable) [18,19].-Unstimulated whole salivary flow (UWSF) was measured at baseline and after the treatment period.-Head and neck high-resolution B-mode sonography using a 9–14 MHz linear transducer (Acuson S2000, Siemens Healthcare, Erlangen, Germany) was performed for all patients in order to assess glandular morphology. To evaluate the glandular stiffness, the glandular shear wave velocity was evaluated applying virtual touch tissue quantification in a technique described in detail previously [20].-Side effects of the inhalative treatment were documented.

### 2.3. Inhalative Agent

Patients received a liposomal inhalation (LipoAerosol, Optima Medical Swiss AG, Zug, Switzerland), containing liposomes, i.e., phospholipids bilayer vesicles, which shape the main constituents of the surfactant film which covers the air/liquid interface on the airways from the lower to the upper respiratory system [12]. LipoAeroso^®^ is a medical device in accordance with Medical Device Directive 93/42/ECC, which obtained CE-marking in 2012 as the first commercially available liposomal inhalation solution and is based on a physiological saline solution with the addition of phospholipid-liposomes made of highly purified lecithin. Patients were instructed in the inhalation therapy and used it three times per day over a period of two months. During each individual inhalation, patients used two ml of the liposomal solution. The mouthpiece was used for the inhalation (Meganeb Nebulizer, Norditalia, Desenzano del Garda, Italy).

### 2.4. Statistical Analysis

Statistical analysis was carried out using version 28.0 of the Statistical Package for Social Sciences software (SPSS, Chicago, IL, USA). Descriptive data are reported as mean ± standard deviation, if not otherwise stated. Normal distribution of variables was tested by using the Kolmogorov-Smirnov test. Paired *t* tests were used for normally distributed variables and Wilcoxon test for not normally distributed variables. *p*-Values of less than 0.05 were considered as statistically significant. Following the intention-to-treat principle, all included patients were included in the final analysis.

## 3. Results

### 3.1. Study Population

A total of 45 patients were included between February 2018 and December 2018. Details on the population are shown in Table 1.

### 3.2. Effect on Subjective Parameters

At baseline, patients rated their xerostomia with a mean score of 30.3 ± 15.3 on the visual analogue scale with a range of 0 to 100. Dysphagia at baseline was rated with a score of 65.6 ± 11.6 in the Anderson Dysphagia Inventory (rather noticeable). Dysphonia at baseline was rated with a mean score of 16.0 ± 19.6 in the Voice Handicap Index (more likely unnoticeable than conspicuous grade of suffering). Quality of life was rated at baseline with a mean score of 53.8 ± 10.3. After the treatment period of two months no statistically significant or clinically relevant changes occurred (Table 2).

The combined ESSPRI score at baseline was 5.0 ± 2.1. After the treatment period this score improved statistically significantly to a mean score of 4.1 ± 2.4 (*p* = 0.012). Looking into the score, this improvement was based on an improvement of the dryness score, which improved from a baseline level of 5.7 ± 2.1 to a level of 4.1 ± 2.4 (*p* < 0.001, Figure 1, Table 3).

### 3.3. Effect on UWSF and Sonographic Parameter

The UWSF and glandular stiffness were evaluated as objective parameters. At baseline, the mean UWSF was 0.65 ± 0.57 mL/5 min, which is above the diagnostic value of 0.5 mL/5 min which is the cut-off for Sjögren’s syndrome. The glandular stiffness, measured with the sonographic evaluation of the shear wave velocity within the parotid and submandibular gland, was 1.89 ± 0.53 m/s and 1.72 ± 0.44 m/s respectively. Neither parameter revealed any statistically significant changes after the treatment period (Table 4).

### 3.4. Therapy Adherence

Nine patients discontinued the inhalation therapy prematurely. Three patients discontinued therapy due to a suspected intolerance towards ingredients within the inhalation solution (soy), two due to worsening of sicca symptoms, two due to irritative cough during inhalation and another two due to fear of infection during inhalation in combination with immunosuppressive therapy. Apart from this, the inhalative treatment was well tolerated and no further side effects were documented.

## 4. Discussion

The current study evaluated the effect of a liposomal inhalative treatment on different laryngeal symptoms in patients with Sjögren’s syndrome in order to evaluate this application technique and also to provide a first impression of the effect in general to enable future case calculations based on the presented results.

Particularly in recent years, several publications have described voice and swallowing disorders in patients with Sjögren’s syndrome [8]. Swallowing disorders are caused by dryness, which impairs the oral and pharyngeal transport of food. Patients often require fluids to support the transport and remove residues. In the most harmless cases, this leads to longer meal times, in extreme cases to malnutrition, but in any case, to a reduction in the quality of life. Voice disorders in patients with Sjögren’s syndrome are described by the changed vibration properties of the laryngeal structures as a result of the hyposalivation. The patients included in this study initially showed comparatively little impairment of voice and swallowing functions based on the results of the Voice Handicap Index and the Anderson Dysphagia Inventory. Other studies on similar parameters reported that 96.3% of the included patients with Sjögren’s syndrome had moderate to severe impairment from their swallowing problems but also voice problems, to a lesser extent (48%). In this study, another score to evaluate dysphonia was applied—the Dysphonia Severity Index (DSI). It was shown that patients suffered from mild (39%), moderate (33%) or even severe (6%) hoarseness. Several other studies made similar findings [9,21,22].

The liposomal inhalation applied in this study contains phospholipids and enables, in addition to the humidification of the respiratory tract, the supplementation of important constituents and the restoration of the surfactant film [12]. In a recent randomized controlled trial, the effect of this liposomal inhalation was compared to an inhalation with physiologic saline solution in newly tracheotomized patients in a double-blind design. It was demonstrated that the proinflammatory Interleukin 6, as well as further objective and subjective parameters, could be significantly improved in patients receiving the liposomal inhalation [12]. A local therapy with liposomal agents has been evaluated in patients with Sjögren’s syndrome before. Different application forms (mouth spray, nose spray, eye spray) have been used to treat xerostomia, rhinitis sicca and keratoconjunctivitis sicca in 73 patients an significant improvement could be achieved [11]. Results from another trial in patients after treatment of head and neck cancer demonstrated that a liposomal spray could improve smell and taste disorders in addition to xerostomia [23].

In the inhalation therapy of liposomes, a few points must be taken into account so that the liposomes can actually develop their effect in the area of the larynx. The particle size of the aerosol generated by the nebulizer is of crucial importance. In the presented study, nebulizers were used that produce a particle size of 5 to 10 μm, which is the ideal size for use in the area of the larynx. Particles larger than 10 μm fade in the nasopharynx, while particles with a size between 3 to 6 μm reach the large- and medium-sized bronchi.

The effect of the liposomal inhalation in our study was mainly demonstrated by the positive effect on the ESSPRI, an established patient-oriented outcome parameter in studies in Sjögren’s syndrome. Other subjective parameters, such as xerostomia, Voice Handicap Index and Anderson Dysphagia Inventory did not reveal any relevant improvement. An explanation for this observation is that the generated particle size was not meant to have much effect on oral complaints such as xerostomia and that the baseline levels of both the Voice Handicap Index and the Anderson Dysphagia Inventory were already within normal values. In contrast, the ESSPRI, which also includes dryness as a general value, showed an improvement that was essentially due to an improvement in the general perceived dryness. Some patients also explicitly stated an improvement in their perceived pulmonary dryness and a previously existing irritable cough. While the minimal clinically important improvement of 1.0 was not reached for the entire score, the dryness item improved by 1.6 points. For the ESSPRI, the patient acceptable symptom state is defined as a score of 5.0 or less. While the ESSPRI at baseline with a mean score of 5.0 was borderline, the dryness item was clearly above this limit with a score of 5.7 on average [24].

The evaluation of glandular stiffness, evaluated with the sonoelastographic method of shear wave velocity (syn. acoustic radiation force impulse imaging, virtual touch tissue quantification) was also included as an objective outcome parameter. Previous studies demonstrated that the measurement of the shear wave velocity is a useful diagnostic tool especially to identify early Sjögren’s syndrome [20]. In addition to these findings, it was also reported that shear wave velocity was suitable for monitoring the effect of local treatments for xerostomia in Sjögren’s syndrome. With the application of shear wave velocity, a decline in the parotid gland stiffness could be measured during an oral treatment with liposomes [25]. However, in the presented results the shear wave velocity of the parotid and submandibular glands was not changed—again this can be explained by the normal baseline values in this cohort based on the finding of Knopf et al. and by the particle size generated by the nebulizer, which was not intended to have much effect within the oral cavity [20].

Some limitations of this study need to be addressed. The design of this study was not controlled and not blinded. This design was chosen with the intention of obtaining a first impression of the effect of a liposomal inhalation on the reported outcome parameters to enable future case calculations in controlled trials. Nevertheless, this means that it cannot be said with certainty whether the effects described are actually due to the inhalation of liposomes or to inhalation in general.

In conclusion, after the inhalative application of liposomes, an improvement of the subjective reported dryness in patients with Sjögren’s syndrome after a treatment period of two months was observed. This was also reflected in the improvement of the ESSPRI and overall the therapy was well tolerated. A first insight into the effect of inhalation therapy on laryngeal symptoms could thus be obtained and at the same time the basis was created on which case calculations can be carried out in the future.

## Figures and Tables

**Figure 1 jcm-11-01081-f001:**
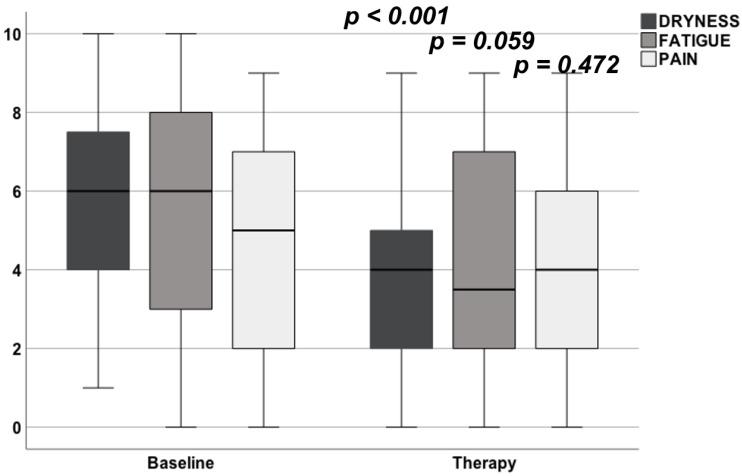
Effect of the liposomal inhalative therapy on the single items of the ESSPRI at baseline and after therapy. The significant improvement of the ESSPRI was mainly based on the improvement of general reported dryness.

**Table 1 jcm-11-01081-t001:** Details of the study population (*n* = 45) (UWSF = unstimulated whole salivary flow, ESSDAI = EULAR SS disease activity index, ESSPRI = EULAR SS patient reported index).

Age (years)	58.2 ± 15.6
Gender (%female)	90
Disease duration (years)	7.1 ± 5.4
Keratoconjunctivitis sicca (% positive)	86
Xerostomia (% positive)	89
Schirmer’s test (% positive)	43
Histopathology (% positive)	71
UWSF (% positive)	96
Antibodies to Ro (SS-A) (% positive)	48
Antibodies to La (SS-B) (% positive)	29
ESSDAI	7.0 ± 8.5
ESSPRI	5.1 ± 2.0

**Table 2 jcm-11-01081-t002:** Xerostomia, dysphagia, dysphonia and quality of life at baseline and after therapy (ADI = Anderson Dysphagia Inventory, VHI = Voice Handicap Index, EORTC QLQ H&N = European Organization for Research and Treatment of Cancer quality of life questionnaire head and neck).

Parameter	Baseline	Therapy	*p*-Value
Xerostomia	30.3 ± 15.3	26.9 ± 15.2	0.066
ADI	65.6 ± 11.6	66.5 ± 12.0	0.583
VHI	16.0 ± 19.6	15.7 ± 15.5	0.614
EORTC QLQ H&N 35	53.8 ± 10.3	50.9 ± 9.2	0.063

**Table 3 jcm-11-01081-t003:** Improvement of the ESSPRI (EULAR SS patient reported index) during the course of the treatment.

Parameter	Baseline	Therapy	*p*-Value
ESSPRI	5.0 ± 2.1	4.1 ± 2.4	0.012
Dryness	5.7 ± 2.1	4.1 ± 2.4	<0.001
Fatigue	5.0 ± 3.0	4.1 ± 3.0	0.059
Pain	4.6 ± 2.4	4.2 ± 2.8	0.472

**Table 4 jcm-11-01081-t004:** Glandular function and glandular stiffness (UWSF = unstimulated whole salivary flow, SWV = shear wave velocity, PG = parotid gland, SMG = submandibular gland).

Parameter	Baseline	Therapy	*p*-Value
UWSF (mL/5 min)	0.65 ± 0.57	0.70 ± 0.64	0.500
SWV PG (m/s)	1.89 ± 0.53	1.89 ± 0.43	0.987
SWV SMG (m/s)	1.72 ± 0.44	1.64 ± 0.40	0.336

## Data Availability

The data presented in this study are available on request from the corresponding author.

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
