# Peer review of "Inhalative Treatment of Laryngitis Sicca in Patients with Sjögren’s Syndrome—A Pilot Study"

_jcm, 2022, doi:10.3390/jcm11041081_

Round 1

Reviewer 1 Report

In this non-randomized uncontrolled pilot clinical trial, Hofauer et al had studied the effect of a two-month inhalative application of a commercially available liposomal moisturizing solution (LipoAerosol, Optima Medical Swiss AG, Switzerland) in 45 patients with Sjögren´s syndrome.

Such liposomal inhalation therapy was evaluated at baseline and after the treatment period using several standardized questionnaires (patient reported indices) and measurements of unstimulated whole salivary flow and sonographic evaluation of glandular stiffness.

Authors have found a significant improvement of the EULAR Sjögren´s syndrome patient reported index (ESSPRI, p=0.012) with subjective improvement of dryness, but no improvement of whole salivary flows and other parameters tested.

My comments:

Interesting study, which may indicate the beneficial subjective effect of a moisturizing solution in the xerostomia of SS patients.

Authors report that nine patients had discontinued the inhalation therapy prematurely, owing to intolerance (including worsening of sicca symptoms). Authors should report the eventual outcome of these patients. 

Minor point:

In the summary: “Forty-five patients could be included in this study” would be more suitably changed to “Forty-five patients were included in this study”

Author Response

Response to reviewer 1:

In this non-randomized uncontrolled pilot clinical trial, Hofauer et al had studied the effect of a two-month inhalative application of a commercially available liposomal moisturizing solution (LipoAerosol, Optima Medical Swiss AG, Switzerland) in 45 patients with Sjögren´s syndrome.

Such liposomal inhalation therapy was evaluated at baseline and after the treatment period using several standardized questionnaires (patient reported indices) and measurements of unstimulated whole salivary flow and sonographic evaluation of glandular stiffness.

Authors have found a significant improvement of the EULAR Sjögren´s syndrome patient reported index (ESSPRI, p=0.012) with subjective improvement of dryness, but no improvement of whole salivary flows and other parameters tested.

Thank you very much for the evaluation of the manuscript.

My comments:

Interesting study, which may indicate the beneficial subjective effect of a moisturizing solution in the xerostomia of SS patients.

Authors report that nine patients had discontinued the inhalation therapy prematurely, owing to intolerance (including worsening of sicca symptoms). Authors should report the eventual outcome of these patients. 

Thank you very much for this important comment. Unfortunately, the symptom scores were not evaluated again in the patients, who returned the inhalation devices prematurely, so that I cannot provide these numbers.

Minor point:

In the summary: “Forty-five patients could be included in this study” would be more suitably changed to “Forty-five patients were included in this study”

Thank you very much, I corrected it according to your suggestion.

Reviewer 2 Report

SUMMARY

This is an open-label pilot study, including 45 patients with Sjögren’s syndrome treated in 2018 for two months with liposomal inhalation therapy, studying laryngeal symptoms, designed to provide data for sample size calculation in further studies.

Population: Sjögren’s syndrome (ACR-EULAR 2016), n=45, exclusion criteria: allergies, patients treated with drugs with anticholinergic effects. Treatment discontinued during the study period: n=9 (20%)
Intervention: liposomal inhalation (LipoAerosol®, 2 ml) three times per day
Comparator: none
Design: open-label, single-arm study, not randomised, patients’ selection process not specified
Results: ESSPRI (patient-reported outcome, PROs): significantly improved from 5.0 to 4.1 (p=0.01), driven by the improvement of the dryness domain (5.7 to 4.1, p=0.001). No difference was observed for other PROs: xerostomia (VAS), ADI (dysphagia), VHI (dysphonia), EORTC QLQ H&N 35 (QoL). Objective outcomes: no improvement for unstimulated salivary flow or glandular stiffness (US).

MAJOR COMMENTS

The study is original as this therapeutic approach targeting the laryngeal area with inhalation has not been described in patients with pSS.

Population
Population sample characteristics are different from usual pSS patients included in pSS studies, with a low prevalence of anti-Ro and anti-Lo and a high prevalence of minor salivary glands biopsy positivity. A possible explanation is that patients were recruited by an HNO Department, to which patients were likely to be sent to perform an MSG biopsy or xerostomia-related symptoms. This could affect the external validity of the study.

Intervention
This product has been shown to be effective in patients with tracheostomy in a RCT published in this journal last year. The authors performed a comparable study (open-label, no control group) with local liposomal treatment in nose/mouth for pSS patients and observed improvements in some outcomes.

Results
An improvement of ESSDAI 0.9 is statistically significant but does not reach minimal clinically important improvement (MCII), defined as 1.0. Patient acceptable symptom state (PASS), defined as 5.0 or less, was already reached at inclusion in the study. ESSPRI is known to be relatively stable over time; however, a recent study showed that 70% of patients present a change of ESSPRI of 1 or more points during two years.

REF:
Eun Hye Park, You-Jung Ha, Eun Ha Kang, Yeong Wook Song, R Hal Scofield, Yun Jong Lee, Baseline disease activity influences subsequent achievement of patient acceptable symptom state in Sjögren’s syndrome, Rheumatology, Volume 60, Issue 6, June 2021, Pages 2714–2724, https://doi.org/10.1093/rheumatology/keaa687

Sandoval-Flores MG, Chan-Campos I, Hernández-Molina G. Factors influencing the EULAR Sjögren's Syndrome Patient-Reported Index in primary Sjögren's syndrome. Clin Exp Rheumatol. 2021 Nov-Dec;39 Suppl 133(6):153-158. Epub 2021 Jun 14. PMID: 34128801.

No improvement after the intervention was observed with other PROs and objective outcomes. This can be explained either by the absence of treatment effect or an insufficient statistical power (driven by the number of patients if the effect size to be observed is small).

As stated by the authors, this study was not designed to prove the effect of liposomal inhalation but to gather data on outcomes distribution, allowing sample size calculation for further studies.

As the authors stated, the main limitations of this study are the absence of a control group and the open-label design. Thus, it is impossible to know if the observed improvement of ESSDAI and dryness domain is due to the intervention (liposomal solution or inhalation), a placebo effect, or by chance (ESSPRI variation). Therefore, I don’t share the authors’ conclusions on a “beneficial effect” of liposomal inhalation and would instead describe the observations made as an improvement of subjective dryness symptoms.

Further studies are definitively needed for this original therapeutic approach, ideally with a randomised controlled design. I suggest adding inclusion criteria in further studies for minimal dryness levels to increase statistical power and reduce the sample size needed in further studies.

MINOR COMMENTS

  1. Figure 1, Figure 2 and Table 3 contain the same information and could be combined to increase conciseness.

Author Response

Response to Reviewer 2

Thank you very much for your detailed evaluation of our manuscript.

SUMMARY

This is an open-label pilot study, including 45 patients with Sjögren’s syndrome treated in 2018 for two months with liposomal inhalation therapy, studying laryngeal symptoms, designed to provide data for sample size calculation in further studies.

Population: Sjögren’s syndrome (ACR-EULAR 2016), n=45, exclusion criteria: allergies, patients treated with drugs with anticholinergic effects. Treatment discontinued during the study period: n=9 (20%)
Intervention: liposomal inhalation (LipoAerosol®, 2 ml) three times per day
Comparator: none
Design: open-label, single-arm study, not randomised, patients’ selection process not specified
Results: ESSPRI (patient-reported outcome, PROs): significantly improved from 5.0 to 4.1 (p=0.01), driven by the improvement of the dryness domain (5.7 to 4.1, p=0.001). No difference was observed for other PROs: xerostomia (VAS), ADI (dysphagia), VHI (dysphonia), EORTC QLQ H&N 35 (QoL). Objective outcomes: no improvement for unstimulated salivary flow or glandular stiffness (US).

MAJOR COMMENTS

The study is original as this therapeutic approach targeting the laryngeal area with inhalation has not been described in patients with pSS.

Population
Population sample characteristics are different from usual pSS patients included in pSS studies, with a low prevalence of anti-Ro and anti-Lo and a high prevalence of minor salivary glands biopsy positivity. A possible explanation is that patients were recruited by an HNO Department, to which patients were likely to be sent to perform an MSG biopsy or xerostomia-related symptoms. This could affect the external validity of the study.

Intervention
This product has been shown to be effective in patients with tracheostomy in a RCT published in this journal last year. The authors performed a comparable study (open-label, no control group) with local liposomal treatment in nose/mouth for pSS patients and observed improvements in some outcomes. 

Results
An improvement of ESSDAI 0.9 is statistically significant but does not reach minimal clinically important improvement (MCII), defined as 1.0. Patient acceptable symptom state (PASS), defined as 5.0 or less, was already reached at inclusion in the study. ESSPRI is known to be relatively stable over time; however, a recent study showed that 70% of patients present a change of ESSPRI of 1 or more points during two years.

REF: 
Eun Hye Park, You-Jung Ha, Eun Ha Kang, Yeong Wook Song, R Hal Scofield, Yun Jong Lee, Baseline disease activity influences subsequent achievement of patient acceptable symptom state in Sjögren’s syndrome, Rheumatology, Volume 60, Issue 6, June 2021, Pages 2714–2724, https://doi.org/10.1093/rheumatology/keaa687

Sandoval-Flores MG, Chan-Campos I, Hernández-Molina G. Factors influencing the EULAR Sjögren's Syndrome Patient-Reported Index in primary Sjögren's syndrome. Clin Exp Rheumatol. 2021 Nov-Dec;39 Suppl 133(6):153-158. Epub 2021 Jun 14. PMID: 34128801.

Thank you very much for this important comment. We included a statement on this into the discussion and also incorporated one of the mentioned studies.

No improvement after the intervention was observed with other PROs and objective outcomes. This can be explained either by the absence of treatment effect or an insufficient statistical power (driven by the number of patients if the effect size to be observed is small).

As stated by the authors, this study was not designed to prove the effect of liposomal inhalation but to gather data on outcomes distribution, allowing sample size calculation for further studies.

As the authors stated, the main limitations of this study are the absence of a control group and the open-label design. Thus, it is impossible to know if the observed improvement of ESSDAI and dryness domain is due to the intervention (liposomal solution or inhalation), a placebo effect, or by chance (ESSPRI variation). Therefore, I don’t share the authors’ conclusions on a “beneficial effect” of liposomal inhalation and would instead describe the observations made as an improvement of subjective dryness symptoms.

Thank you very much, I modified the conclusion accordingly.

Further studies are definitively needed for this original therapeutic approach, ideally with a randomised controlled design. I suggest adding inclusion criteria in further studies for minimal dryness levels to increase statistical power and reduce the sample size needed in further studies.

MINOR COMMENTS

  1. Figure 1, Figure 2 and Table 3 contain the same information and could be combined to increase conciseness.

I agree, therefore I deleted the figure 1, but left table 3 in the manuscript to provide the numbers behind the columns.

Round 2

Reviewer 2 Report

I thank the authors, my comments have been adressed accordingly.

I've no additionnal comment.